# Catalytic transformation of dinitrogen into ammonia and hydrazine by iron-dinitrogen complexes bearing pincer ligand

Shogo Kuriyama[1], Kazuya Arashiba[1], Kazunari Nakajima[1], Yuki Matsuo[2], Hiromasa Tanaka[2], Kazuyuki Ishii[3], Kazunari Yoshizawa[2,4] & Yoshiaki Nishibayashi[1]

Synthesis and reactivity of iron-dinitrogen complexes have been extensively studied, because the iron atom plays an important role in the industrial and biological nitrogen fixation. As a result, iron-catalyzed reduction of molecular dinitrogen into ammonia has recently been achieved. Here we show that an iron-dinitrogen complex bearing an anionic PNP-pincer ligand works as an effective catalyst towards the catalytic nitrogen fixation, where a mixture of ammonia and hydrazine is produced. In the present reaction system, molecular dinitrogen is catalytically and directly converted into hydrazine by using transition metal-dinitrogen complexes as catalysts. Because hydrazine is considered as a key intermediate in the nitrogen fixation in nitrogenase, the findings described in this paper provide an opportunity to elucidate the reaction mechanism in nitrogenase.

[1] Department of Systems Innovation, School of Engineering, The University of Tokyo, Bunkyo-ku, Tokyo 113-8656, Japan. [2] Institute for Materials Chemistry and Engineering and International Research Center for Molecular System, Kyushu University, Nishi-ku, Fukuoka 819-0395, Japan. [3] Institute of Industrial Science, The University of Tokyo, 4-6-1 Komaba, Meguro-ku, Tokyo 153-8505, Japan. [4] Elements Strategy Initiative for Catalysts and Batteries (ESICB), Kyoto University, Nishikyo-ku, Kyoto 615-8520, Japan. Correspondence and requests for materials should be addressed to K.Y. (email: kazunari@ms.ifoc.kyushu-u.ac.jp) or to Y.N. (email: ynishiba@sys.t.u-tokyo.ac.jp).

From a viewpoint of the function of molybdenum and iron atoms in nitrogenase, the development of the catalytic nitrogen fixation by using molybdenum- and iron-dinitrogen complexes as catalysts is one of the most important subjects in chemistry[1]. After the extensive study on the preparation and stoichiometric reactivity of various transition metal-dinitrogen complexes[2–9], the molybdenum-catalyzed nitrogen fixation by using molybdenum-dinitrogen complexes as catalysts under ambient reaction conditions has been achieved by Schrock and co-workers[10,11] and our research group[12–15]. More recently, we have found the most efficient catalytic nitrogen fixation system by using molybdenum-nitride complexes bearing a tridentate triphosphine (PPP = bis(di-tert-butylphosphinoethyl)phenylphosphine) as a ligand, where up to 63 equiv of ammonia were produced based on the catalyst[16].

In addition to the molybdenum-catalyzed nitrogen fixation, the iron-catalyzed reduction of molecular dinitrogen by using iron complexes under mild reaction conditions has recently been achieved because iron-catalyzed nitrogen fixation has also attracted attention from a viewpoint of the industrial nitrogen fixation (the Haber–Bosch process)[17]. In 2012, we found the first successful example of the iron-catalyzed reduction of molecular dinitrogen under ambient reaction conditions, where simple iron complexes such as $[Fe(CO)_5]$ and ferrocene derivatives worked as effective catalysts towards the formation of silylamine as an ammonia equivalent (up to 34 equiv based on the catalyst)[18]. In 2013, Peters and co-workers[19] reported the iron-catalyzed direct reduction of molecular dinitriogen into ammonia under mild reaction conditions (1 atm at −78 °C), where a sophisticated iron-dinitrogen complex bearing a triphosphine-borane as a ligand worked as a catalyst (up to 7 equiv of ammonia based on the catalyst). More recently, Peters and co-workers[20,21] have found the other iron-catalyzed nitrogen fixation system by using iron-complexes bearing a triphosphinealkyl ligand and two cyclic carbene ligands, where up to 4.6 equiv and 3.4 equiv of ammonia were produced based on the catalyst, respectively. However, the detailed reaction pathway has not yet been reported in all the iron-catalyzed nitrogen fixation systems[22].

Based on our previous findings of the unique catalytic activity of molybdenum complexes bearing mer-tridentate ligands such as PNP′-pincer ligands (PNP′ = 2,6-bis(di-tert-butylphosphino-methyl)pyridine)[12–15] and PPP[16] ligand, we have designed iron-dinitrogen complexes bearing PNP′-pincer and PPP ligands as catalysts towards the iron-catalyzed nitrogen fixation. Although we have not yet succeeded in preparing the corresponding iron-dinitrogen complexes bearing PNP′-pincer and PPP ligands, we

have been successful to prepare a similar iron-dinitrogen complex bearing an anionic PNP-pincer ligand (PNP = 2,5-bis(di-tert-butylphosphinomethyl)pyrrolide)[23–25] ($[Fe(N_2)(PNP)]$: **1**). As a result, an iron-dinitrogen complex as well as its precursors iron-hydride and -methyl complexes have been found to work as effective catalysts towards the catalytic nitrogen fixation under mild reaction conditions. Interestingly, a mixture of ammonia and hydrazine was obtained as nitrogenous products in the present reaction system. Herein, we report the catalytic reduction of molecular dinitrogen into ammonia and hydrazine by using the iron complexes bearing an anionic PNP-pincer ligand as catalysts.

## Results

**Preparation and characterization of iron complexes.** The reaction of $[FeCl_2(thf)_{1.5}]$ with lithium 2,5-bis(di-tert-butylpho-sphinomethyl)pyrrolide, generated from 2,5-bis(di-tert-butylpho-sphinomethyl)pyrrole and $^nBuLi$, in toluene at room temperature for 14 h gave an iron-chloride complex bearing PNP ligand, $[FeCl(PNP)]$, (**2**) in 85% yield (Fig. 1). Reduction of **2** with 1.1 equiv of $KC_8$ as a reductant in tetrahydrofuran (THF) at room temperature for 13 h under an atmospheric pressure of dinitrogen afforded a paramagnetic iron(I)-dinitrogen complex **1** in 68% yield. Molecular structures of **1** and **2** were confirmed by X-ray analysis. ORTEP drawings of **1** and **2** are shown in Fig. 2a,b. Crystal structures of both **1** and **2** have a distorted square-planar geometry around the iron atom (the geometry index $\tau_4 = 0.13$ and $\tau_4 = 0.11$, respectively), where $\tau_4 = 0.00$ for a perfect square-planar and $\tau_4 = 1.00$ for a tetrahedral geometry[26]. A dinitrogen ligand coordinates to the iron atom in a terminal fashion with the Fe–N distance of 1.764(2) Å and the N–N distance of 1.134(2) Å. To our knowledge, only a few examples of square-planar iron complexes bearing a terminal dinitrogen ligand, except for iron-dinitrogen complexes bearing a 2,6-bis(imine)pyridine ligand[27–29], have been reported until now.

The infrared (IR) spectrum of **1** in solid state (KBr) shows a strong $\nu_{NN}$ band at 1,964 cm$^{-1}$ assignable to the terminal dinitrogen ligand. The complex **1** in a THF solution shows a $\nu_{NN}$ band at 1,966 cm$^{-1}$, which is similar to that of **1** in a solid state. Cyclic voltammetry of **1** in THF with $[N^nBu_4]PF_6$ as a supporting electrolyte revealed an irreversible reduction at −2.9 V versus ferrocene$^{0/+}$ and an irreversible oxidation at −0.9 V (Supplementary Fig. 1). The reduction and oxidation can be assignable to Fe(0/I) and Fe(I/II) respectively. Electron para-magnetic resonance (EPR) measurements were carried out at 10

**Figure 1 | Synthesis and reactivity of iron complexes.** The reaction of iron(II) chloride with PNP-Li afforded **2**. Reduction of **2** under $N_2$ atmosphere gave **1**. Reactions of **2** with KBHEt$_3$ and MeMgCl afforded **3** and **4**, respectively. The complexes **3** and **4** were converted into **1** on protonation and reduction under $N_2$ atmosphere.

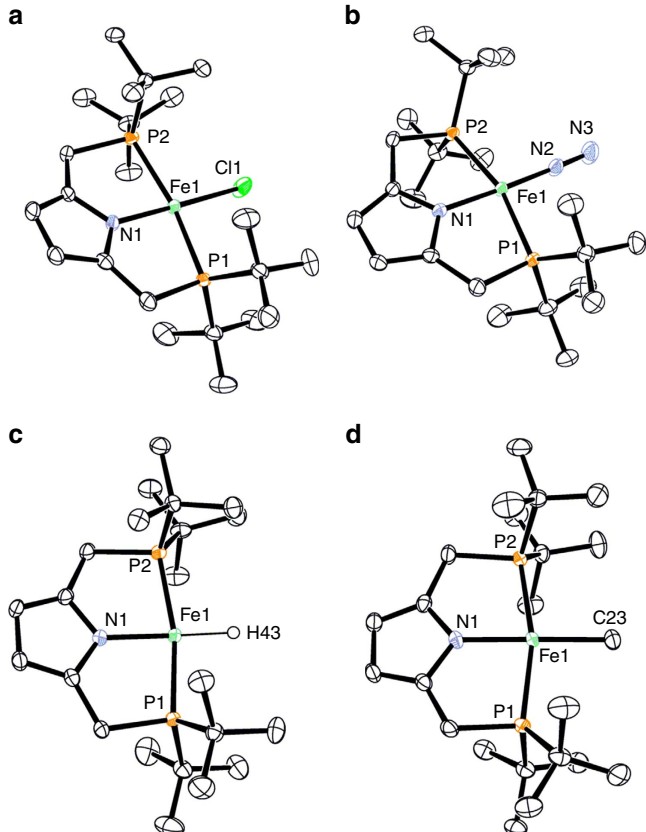

**Figure 2 | ORTEP drawings of the iron complexes.** (**a**) chloride complex **2**, (**b**) dinitrogen complex **1**, (**c**) hydride complex **3** and (**d**) methyl complex **4**. Thermal ellipsoids are shown at the 50% level. Hydrogen atoms except for H43 in **3** are omitted for clarity.

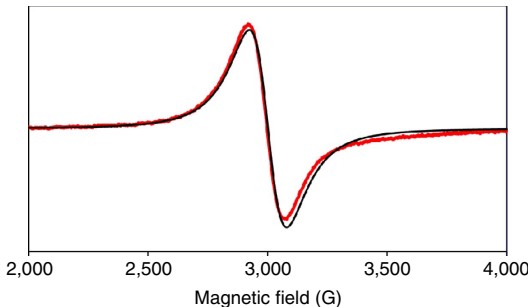

**Figure 3 | X-band EPR spectra of 1.** The spectrum collected at room temperature in a toluene solution at a microwave frequency 9.44 GHz (red) and the simulated EPR spectrum of **1** (black).

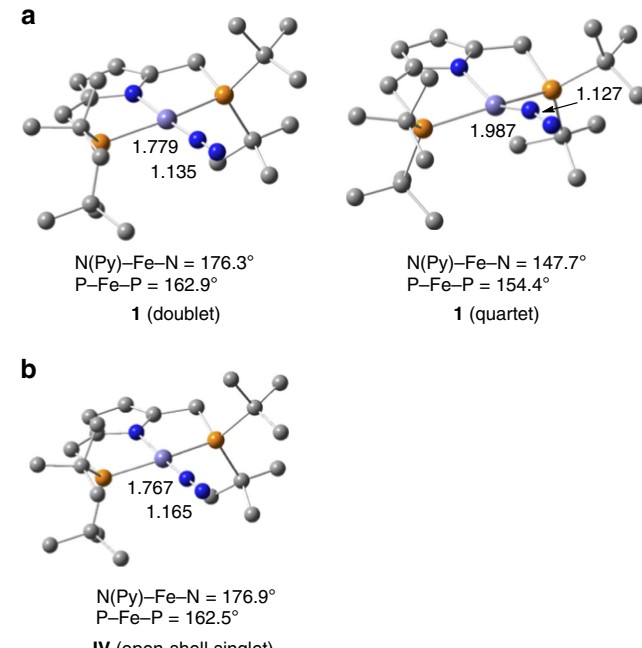

**Figure 4 | Optimized structures of 1 and IV.** (**a**) The structures of **1** in the doublet and quartet states and (**b**) the structure of **IV** in the open-shell singlet state. Bond distances are presented in Å. Hydrogen atoms are omitted for clarity.

and 296 K in toluene to investigate the spin state of the complex **1**. The complex **1** shows a typical single EPR signal at room temperature, which is attributable to an $S = 1/2$ system (Fig. 3). The g value ($g = 2.25$) of **1** is largely deviated from that (2.0023) of a free electron. The width between extreme slope (140 G) of this EPR signal is much broader than that of conventional organic radicals, and therefore, any hyperfine structures could not be seen. The deviation of the g value and the broad bandwidth are characteristic features of EPR of metallocomplexes. In the EPR spectrum of **1** at 10 K (Supplementary Fig. 2), anisotropic EPR signals are observed at around $g = 2$, but no EPR signal is seen at around $g = 4$, suggesting $S = 1/2$. Reproducible EPR signals at $g = 2.6$ and 2.2 are attributable to the EPR of **1**, which are similar to the previous EPR spectra observed for the low-spin iron(I)

complexes with a square-planar geometry[30]. The complex **1** has a solution magnetic moment of $3.0 \pm 0.2\ \mu_B$ at 298 K. The measured magnetic moment is larger than spin-only value for an $S = 1/2$ spin state ($1.73\ \mu_B$), but still within the range of the reported low-spin square-planar iron(I) compounds[30,31]. We consider that the large shift of the magnetic moment of **1** may be a result of the spin-orbit coupling.

Density functional theory (DFT) calculations at the B3LYP-D3 level of theory[32] have been carried out to discuss the ground spin state structure of **1**. Optimized structures of **1** in the doublet and quartet states are depicted in Fig. 4, together with their selected geometric parameters. The result of the B3LYP-D3 calculations indicates that the ground spin state of **1** is doublet and the quartet state lies above 9.2 kcal mol$^{-1}$. The Fe–N$_2$ and N–N distances are calculated to be 1.779 and 1.135 Å in the doublet state and 1.987 and 1.127 Å in the quartet state, respectively, the former of which are close to those in the crystal structure of **1** shown in Fig. 2a. The geometry index $\tau_4$ in the doublet state (0.15) well reproduces a slightly distorted square-planar geometry around the iron atom in the crystal structure, while the quartet state structure has a larger value of $\tau_4$ (0.41). All the results strongly support the experimental finding that the ground spin state of **1** is doublet.

Reactions of **2** with KBHEt$_3$ in THF and MeMgCl in Et$_2$O at room temperature for 1 h afforded paramagnetic iron(II)-hydride and -methyl complexes, [FeH(PNP)] (**3**) and [FeMe(PNP)] (**4**), in 62 and 81% yields, respectively (Fig. 1). Molecular structures of **3** and **4** were confirmed by X-ray analysis. ORTEP drawings of **3** and **4** are shown in Fig. 2c,d. Crystal structures of both **3** and **4** have a distorted square-planar geometry around the iron atom ($\tau_4 = 0.11$ and $\tau_4 = 0.12$, respectively). The iron(II) complexes **2**–**4** have solution magnetic moments of $3.7 \pm 0.2$, $3.6 \pm 0.2$ and $3.8 \pm 0.2\ \mu_B$ at 296 K, respectively. The measured magnetic moments are larger than spin-only value for an $S = 1$ spin state ($2.83\ \mu_B$), but still within the range of the reported intermediate-spin square-planar iron(II) compounds[33–35]. When the magnetic moments and the square-planar structures are taken into account,

the complexes **2**–**4** can be assigned as intermediate-spin $S = 1$ states. We consider that the spin-orbit coupling may contribute the large magnetic moments of **2**–**4**.

**Reactivity of iron complexes**. At first, the catalytic reaction was carried out by using **1** as a catalyst under our reaction conditions, where $CoCp_2$ and 2,6-lutidinium trifluoromethanesulfonate ([LutH]OTf) were used as a reductant and a proton source at room temperature[12–15]. However, no formation of ammonia was observed at all. Then, we investigated the catalytic reaction under the reaction conditions previously reported by Peters and co-workers[19–22]. Typical results are shown in Table 1. The reaction of an atmospheric pressure of dinitrogen with $KC_8$ (40 equiv to **1**) as a reductant and $[H(OEt_2)_2]BAr^F_4$ (38 equiv to **1**; $Ar^F = 3$, 5-bis(trifluoromethyl)phenyl) as a proton source in the presence of **1** as a catalyst in $Et_2O$ at −78 °C for 1 h gave 4.4 equiv of ammonia and 0.2 equiv of hydrazine based on the iron atom of the catalyst, respectively (Table 1, run 1). When the reaction was carried out at room temperature, the formation of ammonia and hydrazine was not observed, but only molecular dihydrogen (5.2 equiv) was produced. The use of larger amounts of both reductant and proton source increased the amounts of both ammonia and hydrazine, up to 10.9 equiv and 1.6 equiv, respectively (Table 1, runs 2 and 3). The largest amounts of ammonia and hydrazine (14.3 equiv of ammonia and 1.8 equiv of hydrazine) were obtained by using 200 equiv of $KC_8$ and 184 equiv of $[H(OEt_2)_2]BAr^F_4$ under the same reaction conditions (Table 1, run 4). Separately, we confirmed the direct conversion of molecular dinitrogen into ammonia and hydrazine using $^{15}N_2$ gas instead of $^{14}N_2$ gas. After the catalytic reaction, we could not identify any iron complexes and only the formation of free PNP-H was observed by nuclear magnetic resonance (NMR).

The ratio of ammonia to hydrazine depends on the nature of solvents. When THF was used in place of $Et_2O$, hydrazine was produced as a major product together with ammonia as a minor product based on the fixed N atom, where up to 2.4 equiv of hydrazine and 2.9 equiv of ammonia were produced based on the catalyst (Table 1, runs 5 and 6). Iron- and other transition metal-dinitrogen complexes have been reported to produce a stoichiometric amount of hydrazine on treatment of acids and reductants[36–39]. This result shows that the dinitrogen complex **1** works as a catalyst for the formation of hydrazine directly from dinitrogen. The use of much larger amounts of both reductant and proton source did not increase the amounts of both hydrazine and ammonia (Table 1, run 7). The formation of ammonia and hydrazine was not observed from the reaction in $Et_2O$ at −78 °C in the absence of iron-complexes as catalysts (Table 1, runs 8 and 9).

Interestingly, iron-hydride and -methyl complexes **3** and **4** also worked as effective catalysts under the same reaction conditions, where 3.0 equiv and 3.7 equiv of ammonia were produced based on the iron atom of the catalyst, respectively (Table 1, runs 10 and 11)[19,20,40]. We consider that the unique reactivity of iron-hydride complex **3** provides useful information to consider the reaction mechanism of nitrogenase because iron-hydride complexes are reported to play an important role as a key reactive intermediate in the catalytic cycle of nitrogenase[1,41]. Separately, we confirmed the protonation of **3** and **4** with 1 equiv of $[H(OEt_2)_2]BAr^F_4$ in $Et_2O$ at room temperature and then the addition of 1 equiv of $KC_8$ under $N_2$ (1 atm) gave **1** together with molecular dihydrogen and methane, respectively. These results indicate that **3** and **4** are easily converted into **1** under the catalytic reaction conditions. Schrock and a coworker previously reported a similar phenomenon that protonation and reduction of a molybdenum-hydride complex bearing a triamideamine ligand under $N_2$ (1 atm) gave the corresponding molybdenum-dinitrogen complex, which was worked as a catalyst for the formation of ammonia from molecular dinitrogen[42]. After the submission of the manuscript, Peters and co-workers[43] have reported that an iron-hydride complex bearing a triphosphine-borane ligand was identified to be catalytically competent when it was solubilized, and also was identified to be a catalyst resting state, although Peters and co-workers reported that iron-hydride complexes did not work as catalysts in the previous papers[19,20,40]. On the other hand, only a stoichiometric amount of ammonia was formed when **2** was used as a catalyst (Table 1, run 12).

---

**Table 1 | Iron-catalyzed reduction of dinitrogen to ammonia and hydrazine\*.**

$$N_2 \text{ (1 atm)} + KC_8 \text{ (40 equiv)} + [H(OEt_2)_2]BAr^F_4 \text{ (38 equiv)} \xrightarrow[\text{solvent} \atop -78\,°C,\ 1\,h]{\text{catalyst}} NH_3 + NH_2NH_2$$

| Run | Catalyst | Solvent | NH₃(equiv)† | NH₂NH₂(equiv)† | fixed N atom (equiv)‡ |
|---|---|---|---|---|---|
| 1§ | **1** | $Et_2O$ | 4.4 ± 0.2 | 0.2 ± 0.2 | 4.8 |
| 2‖ | **1** | $Et_2O$ | 6.7 | 0.8 | 8.3 |
| 3¶ | **1** | $Et_2O$ | 10.9 ± 0.4 | 1.6 ± 0.2 | 14.1 |
| 4# | **1** | $Et_2O$ | 14.3 ± 0.4 | 1.8 ± 0.2 | 17.9 |
| 5 | **1** | THF | 1.9 ± 0.4 | 1.4 ± 0.7 | 4.7 |
| 6‖ | **1** | THF | 2.9 ± 0.2 | 2.4 ± 0.1 | 7.7 |
| 7¶ | **1** | THF | 1.6 | 0.8 | 3.2 |
| 8 | —** | $Et_2O$ | 0 | 0 | 0 |
| 9 | PNP-H†† | $Et_2O$ | <0.1 | 0 | 0 |
| 10 | **3** | $Et_2O$ | 3.0 ± 0.9 | 0.1 ± 0.1 | 3.2 |
| 11 | **4** | $Et_2O$ | 3.7 ± 0.5 | <0.1 | 3.7 |
| 12 | **2** | $Et_2O$ | 1.1 ± 0.6 | 0 | 1.1 |
| 13 | **5** | $Et_2O$ | 2.6 ± 0.2 | <0.1 | 2.7 |

\*A mixture of a catalyst (0.010 mmol), $KC_8$ (0.40 mmol, 40 equiv based on the catalyst), and $[H(OEt_2)_2]BAr^F_4$ (0.38 mmol, 38 equiv based on the catalyst) was stirred in solvent at −78 °C for 1 h under 1 atm of $N_2$ and then at room temperature for 20 min.
†Equiv based on the iron atom of a catalyst. Average of multiple runs (>2 times) are shown unless otherwise stated.
‡Fixed N atom (equiv) = [NH₃ (equiv)] + 2[NH₂NH₂ (equiv)]. Equiv based on the iron atom of a catalyst.
§Average of 5 runs are shown.
‖80 equiv of $KC_8$ and 76 equiv of $[H(OEt_2)_2]BAr^F_4$ were used.
¶160 equiv of $KC_8$ and 152 equiv of $[H(OEt_2)_2]BAr^F_4$ were used.
#200 equiv of $KC_8$ and 184 equiv of $[H(OEt_2)_2]BAr^F_4$ were used.
\*\*In the absence of a catalyst.
††PNP-H ligand (0.010 mmol) was used as a catalyst.

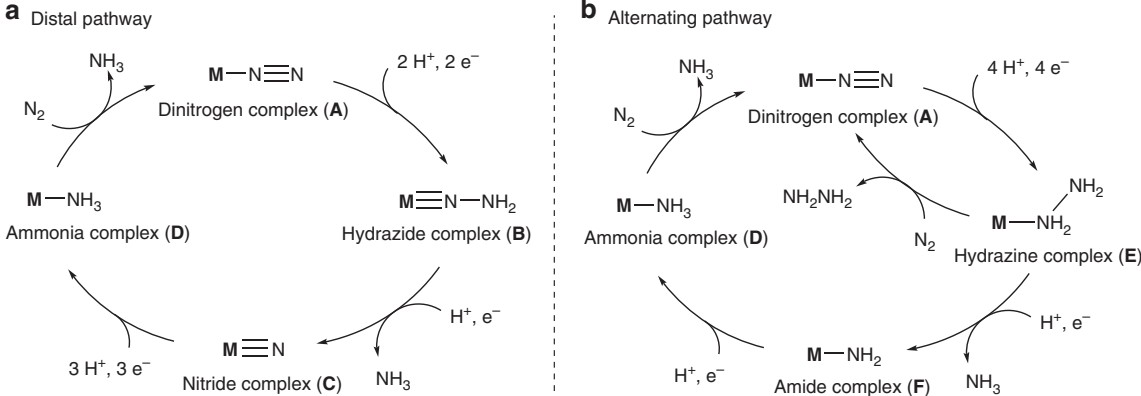

**Figure 5 | Reaction pathway for formation of ammonia and hydrazine from molecular dinitrogen by using transition metal-dinitrogen complexes as catalysts.** (**a**) Distal pathway and (**b**) alternating pathway.

The formation of both ammonia and hydrazine is in sharp contrast to that in the Peters' reaction systems[19–22], where no formation of hydrazine was observed at all when iron-dinitrogen complexes bearing triphosphine-borane and -alkyl ligands, and two cyclic carbene ligands were used as catalysts. Separately, we confirmed that the partial reduction[7,44–47] of hydrazine into ammonia in the presence of a catalytic amount of **4** proceeded in Et$_2$O at −78 °C for 1 h (Supplementary Table 6). However, the use of THF in place of Et$_2$O relatively inhibited the partial reduction of hydrazine into ammonia under the same reaction conditions. These results indicate that some iron-hydrazine complexes may be involved as key reactive intermediates in the transformation of hydrazine into ammonia. We consider that the result described in the present manuscript provides useful information on the elucidation of the reaction mechanism in nitrogenase because hydrazine may be formed as a key reactive intermediate in the biological nitrogen fixation[1].

**Discussion on the catalytic reaction pathway.** Based on the results of experimental and DFT calculations on the molybdenum-catalyzed nitrogen fixation under mild reaction conditions, we proposed that the transformation of molecular dinitrogen into ammonia under mild reaction conditions proceeds via hydrazide and nitride complexes (**B** and **C**, respectively) as key reactive intermediates as shown in Fig. 5a as a distal pathway[12–16]. However, the formation of hydrazine is not possible from a distal pathway. To explain the direct formation of hydrazine from molecular dinitrogen in the present iron system, we now propose an alternating pathway where the catalytic reaction proceeds via hydrazine complex (**E**) as a key reactive intermediate as shown in Fig. 5b[19,22]. In fact, the alternating pathway has been proposed for the stoichiometric formation of ammonia and hydrazine from iron-dinitrogen complexes based on the reactivity of isolated and generated intermediates[36,37]. In this alternating reaction pathway, dinitrogen complex (**A**) is converted into hydrazine complex (**E**) via sequential protonation and reduction. Hydrazine is formed by the ligand exchange of the coordinated hydrazine for molecular dinitrogen to regenerate the starting complex **A**. On the other hand, further protonation and reduction of hydrazine complex **E** give ammonia together with amide complex (**F**). Then, amide complex **F** is transformed into ammonia complex (**D**). Finally, the ligand exchange of the coordinated ammonia for molecular dinitrogen regenerates the starting complex **A**. As presented in the previous section, the ratio of ammonia to hydrazine depends on the nature of solvents. The ligand exchange of the coordinated hydrazine for molecular dinitrogen might proceed more smoothly in THF as solvent to give hydrazine as a major product. At the

present stage, however, we can not exclude the possibility of ammonia formation via the distal reaction pathway. In addition, reduction of dinitrogen via a hybrid of the distal and the alternating pathways, where the hydrazide complex **B** was converted into hydrazine complex **E** on protonation and reduction, is also possible[48].

To obtain information on reactive species in the present catalytic reaction by using **1**, we carried out the protonation of **1** under the following reaction conditions. The protonation of iron-dinitrogen complex **1** with 1 equiv of [H(OEt$_2$)$_2$]BAr$^F_4$ in Et$_2$O at room temperature for 10 min gave the corresponding protonated complex (**5**) in 90% yield (Fig. 6). A $\nu_{NN}$ peak at 2,034 cm$^{-1}$ assignable to the terminal dinitrogen ligand appeared at the IR spectrum of the protonated complex in solid state (KBr) although no $\nu_{NH}$ peak was observed. Based on the experimental result, we characterized **5** as iron-dinitrogen complex including the protonated pyrrole ring of PNP ligand. The result reveals that the first protonation at **1** may occur not at the coordinated dinitrogen ligand but at the pyrrole moiety in **1** (ref. 49). When we attempted to reduce **5** with 1 equiv of KC$_8$ as a reductant in Et$_2$O at room temperature for 10 min, a mixture of **1** and free PNP-H was observed in 38 and 27% yields by $^1$H NMR, respectively, together with the formation of **3** in 3% yield. Separately, we confirmed that the protonated iron-dinitrogen complex **5** has also a slightly lower catalytic activity towards the nitrogen fixation than **1** (Table 1, run 13). These experimental results indicate that the protonated iron-dinitrogen complex **5** is considered to be one of deactivated species in the present catalytic reaction.

**DFT calculations on the reactivity of iron-dinitrogen complexes.** To get further information on the reaction pathway, we have carried out DFT calculations on the first protonation of **1** with H$^+$(OEt$_2$)$_2$, according to the experimental result shown in Fig. 6.

**Figure 6 | Reactivity of iron-dinitrogen complex 1.** Protonation of **1** with [H(OEt$_2$)$_2$]BAr$^F_4$ occurred at the pyrrole ring to give **5**.

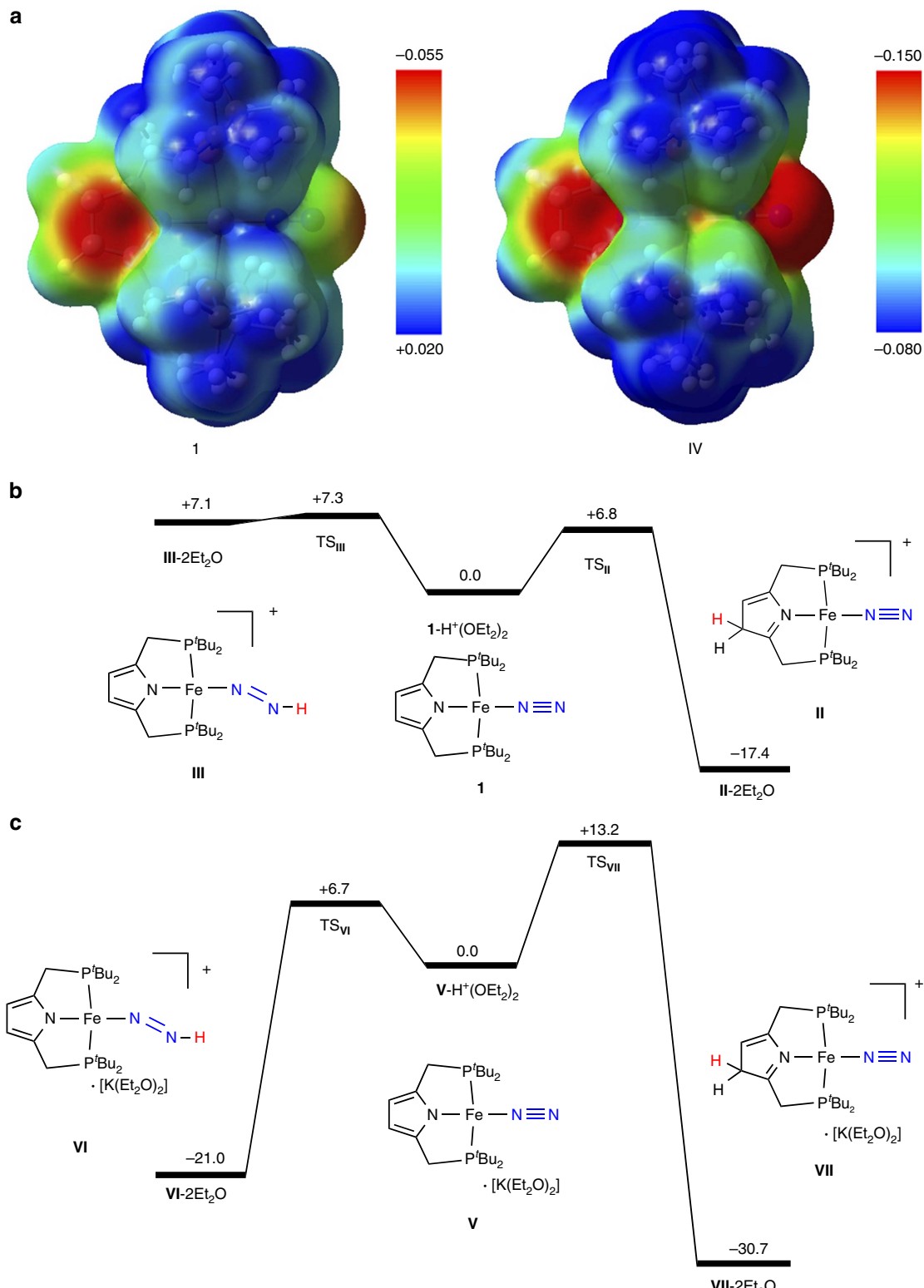

**Figure 7 | Possible reaction pathways.** (**a**) ESP maps (with the scale in atomic units) of **1** (left) and **IV** (right) on the 0.002 electrons/bohr$^3$ isodensity surface, where reddish and bluish surfaces represent electron-rich and -poor regions, respectively. (**b**) Energy profiles of proton transfer from H$^+$(OEt$_2$)$_2$ to the pyrrole ring of the PNP ligand (center to right) and the N$_2$ ligand (center to left) in **1** and (**c**) those in **V**. Gibbs free energy changes (ΔGs) at 195 K are presented in kcal mol$^{-1}$.

The Natural Population Analysis (NPA) charges[50] on the N$_2$ ligand and the pyrrole moiety consisting of NC$_4$H$_2$ were calculated to be −0.20 and −0.59, respectively. Thus, the anionic PNP ligand in **1** should be considered as a possible site for the protonation by H$^+$(OEt$_2$)$_2$. Figure 7a shows an

electrostatic potential (ESP) map of **1**, where reddish region in the mapped isosurface strongly attracts a positively charged species. The ESP map indicates that both the pyrrole moiety and the distal N atom of the N$_2$ ligand are likely to accept a proton. Figure 7b describes energy profiles of proton transfer from

$H^+(OEt_2)_2$ to the pyrrole moiety of PNP (**1** to **II** via **TS$_{II}$**) and the $N_2$ ligand (**1** to **III** via **TS$_{III}$**). The protonation reactions require very low-activation energies, 6.8 kcal mol$^{-1}$ for PNP and 7.3 kcal mol$^{-1}$ for $N_2$, both of which could be low enough to overcome even at 195 K. On the other hand, the protonation of a β-C atom of the pyrrole ring is highly exergonic by 17.4 kcal mol$^{-1}$, while that of the $N_2$ ligand is endergonic by 7.1 kcal mol$^{-1}$. The shifts of $\nu_{NN}$ on protonation are $+83$ cm$^{-1}$ for **II** and $-302$ cm$^{-1}$ for **III**. The blue-shift of **II** is consistent with the observed shift ($+70$ cm$^{-1}$) shown in Fig. 6, which can be ascribed to the protonation of the pyrrole ring of the PNP ligand[49]. All the results indicate that the protonation of **1** is not suitable for the formation of N–H bond of the coordinated $N_2$ ligand.

As described in the previous sections, the protonation of the coordinated dinitrogen ligand may not occur at **1**. Based on the experimental result, we consider that reduction of **1** into the corresponding iron(0)-dinitrogen complexes precedes the protonation on the coordinated dinitrogen ligand of **1** in the catalytic reaction. In fact, Peters and co-workers[19–22,51] reported that anionic low-valent dinitrogen complexes played essential roles in the catalytic activity for reduction of dinitrogen. Based on this assumption, we attempted to reduce **1** and **2** with an excess amount of KC$_8$, K, Na and sodium naphtalenide as reductants in the presence or absence of crown ethers and cryptands under $N_2$ (1 atm). The reaction of **2** with an excess amount of Na sand (5 equiv) in the presence of 1 equiv of 15-crown-5 in THF at room temperature for 15 h under an atmospheric pressure of dinitrogen gave **1** as a main product based on the IR measurement. However, recrystallization of the crude mixture afforded a few brown crystals. The preliminary result of X-ray analysis indicates the formation of an iron(0)-dinitrogen complex [Fe(PNP)(N$_2$)][Na(15-crown-5)]. The obtained crystals showed a $\nu_{NN}$ peak at 1,831 cm$^{-1}$ at the IR spectrum (KBr). The lower IR absorbance assignable to the terminal dinitrogen ligand indicates the possibility of formation of the reduced iron(0)-dinitrogen complex in the reaction mixture. Thus, based on the experimental result, we can consider the formation of reduced iron(0)-dinitrogen complexes under the catalytic reaction conditions.

To consider a reaction pathway that reduction of **1** precedes the protonation, we optimized the structure of an anion complex of **1**, [Fe(PNP)(N$_2$)]$^-$ (**IV**) (Fig. 4b). The ground spin state of **IV** is an open-shell singlet, where the Mulliken spin densities are assigned to Fe ($+0.66$) and $N_2$ ($-0.57$). The closed-shell singlet and triplet states lies 2.3 and 5.1 kcal mol$^{-1}$ above the open-shell singlet state, respectively. The N–N stretching in the open-shell singlet state of **IV** is calculated to be 1859 cm$^{-1}$, which is close to the experimental value (1,831 cm$^{-1}$). The calculated results would suggest the formation of anionic Fe(0)-dinitrogen complex **IV** from **1** in the presence of a suitable reductant. It is notable that the reduction of **1** significantly activates the N≡N bond of the $N_2$ ligand. Reduction elongates the N–N distance by 0.030 Å (1.135 Å → 1.165 Å) and increases negative charge on $N_2$ by 0.26 ($-0.20 \rightarrow -0.46$), while a subtle change in the gross charge is observed in the pyrrole moiety ($-0.59 \rightarrow -0.63$). As shown in Fig. 7a, an ESP map of **IV** indicates that the $N_2$ ligand in the anionic iron complex is expected to attract a proton more strongly compared with that in neutral complex **1**.

We next examined the protonation of the $N_2$ ligand and pyrrole moiety in **IV**, where a solvated potassium ion, $K^+(OEt_2)_2$ is adopted as a counter cation of **IV**. In the lowest-energy structure of [Fe(PNP)N$_2$][K(OEt$_2$)$_2$] (**V**), $K^+(OEt_2)_2$ is located in the vicinity of the pyrrole moiety of the anionic iron complex (see more detail in the Supplementary Material). As shown in Fig. 7c, protonation of the $N_2$ ligand in **V** yielding a diazenide complex **VI** via **TS$_{VI}$** proceeds in a highly exergonic way ($\Delta G = -21.0$

kcal mol$^{-1}$) with a low-activation energy of 6.7 kcal mol$^{-1}$. On the other hand, the positively charged $K^+(OEt_2)_2$ is likely to prevent attack of $H^+(OEt_2)_2$ to the pyrrole moiety. Protonation of a β-C atom of the pyrrole ring in **V** yielding **VII** via **TS$_{VII}$** is also exergonic by 30.7 kcal mol$^{-1}$ and requires an activation energy of 13.2 kcal mol$^{-1}$, which is much higher than that for the protonation of $N_2$. These results indicate that the reduction of **1** would provide a great advantage to the protonation of the $N_2$ ligand, which is the first step towards the transformation of $N_2$ into NH$_3$. Further experimental and theoretical studies are necessary to elucidate the detailed reaction pathway.

## Discussion

We have found that newly designed and prepared iron-dinitrogen, -hydride and –methyl complexes bearing an anionic PNP-pincer ligand worked as effective catalysts towards the iron-catalyzed reduction of molecular dinitrogen into ammonia and hydrazine under mild reaction conditions. Previously Shilov and co-workers[52,53] reported that the reaction of dinitrogen with reductant in protic media in the presence of a catalytic amount of a molybdenum complex afforded hydrazine together with a small amount of ammonia. In this Shilov system, however, the detailed mechanism and the catalytically relevant species such as a dinitrogen complex have remained unclear[52,53]. In the present reaction system, we have achieved the first and clear-cut example of the catalytic direct formation of hydrazine from molecular dinitrogen under mild reaction conditions by using the well-defined transition metal-dinitrogen complexes as catalysts. We consider that the new findings described in this paper provide a new opportunity not only to design and develop more effective nitrogen fixations under mild reaction conditions by using transition metal-dinitrogen complexes as catalysts but also to elucidate the reaction mechanism in nitrogenase.

## Methods

**Preparation of [Fe(N$_2$)(PNP)] (1).** A suspension of **2** (142 mg, 0.300 mmol) and KC$_8$ (44.4 mg, 0.328 mmol) in THF (6 ml) was stirred at room temperature for 13 h under N$_2$ (1 atm). The resultant dark red suspension was concentrated *in vacuo*. To the residue was added hexane (5 ml), and the solution was filtered through Celite, and the filter cake was washed with hexane (2 ml, 5 times). The combined filtrate was concentrated to ca. 3 ml and the solution was kept at –17 °C to give **1** as red crystals, which were collected by decantation, washed with a small amount of cold pentane and dried *in vacuo* (95.4 mg, 0.205 mmol, 68%). $^1$H NMR (C$_6$D$_6$) δ 3.3, $-1.9$, $-15.1$. Magnetic susceptibility (Evans' method)[54]: $\mu_{eff} = 3.0 \pm 0.2$ $\mu_B$ in C$_6$D$_6$ at 296 K. IR (KBr, cm$^{-1}$) 1,964 (s, $\nu_{NN}$). IR (THF, cm$^{-1}$) 1,966 (s, $\nu_{NN}$). Anal. Calcd. for C$_{22}$H$_{42}$FeN$_3$P$_2$: C, 56.66; H, 9.08; N, 9.01. Found: C, 56.88; H, 9.10; N, 8.58.

**Catalytic reduction of dinitrogen to ammonia.** The catalytic reduction of molecular dinitrogen into ammonia and hydrazine was carried out according to a method similar to Peters' procedure[19–22]. A typical experimental procedure using **1** is described below. In a 50-ml Schlenk were placed **1** (4.6 mg, 0.010 mmol), [H(OEt$_2$)$_2$]BAr$^F_4$ (385 mg, 0.380 mmol), and KC$_8$ (54.2 mg, 0.401 mmol). After the mixture was cooled at $-196$ °C, Et$_2$O (5 ml) was added to the mixture by trap-to-trap distillation. The Schlenk flask was warmed to $-78$ °C and then was filled with N$_2$ (1 atm). After stirring for 1 h at $-78$ °C, the mixture was warmed to room temperature and further stirred at room temperature for 20 min. The amount of dihydrogen of the catalytic reaction was determined by gas chromatography analysis. The reaction mixture was evaporated under reduced pressure, and the distillate was trapped in dilute H$_2$SO$_4$ solution (0.5 M, 10 ml). Potassium hydroxide aqueous solution (30 wt%; 5 ml) was added to the residue, and the mixture was distilled into another dilute H$_2$SO$_4$ solution (0.5 M, 10 ml). The amount of NH$_3$ present in each of the H$_2$SO$_4$ solutions was determined by the indophenol method[55]. The amount of NH$_2$NH$_2$ present in each of the H$_2$SO$_4$ solutions was determined by the *p*-(dimethylamino)benzaldehyde method[56].

**Data availability.** Detailed experimental procedures, characterization of compounds, Cartesian coordinates and the computational details can be found in the Supplementary Figs 1–11, Supplementary Tables 1–23 and Supplementary Methods. The X-ray crystallographic coordinates for structures reported in this article have been deposited at the Cambridge Crystallographic Data Centre

(CCDC), under deposition number CCDC-1447087 (**1**), 1447088 (**2**), 1447089 (**3**) and 1447090 (**4**). These data can be obtained free of charge from the Cambridge Crystallographic Data Centre via www.ccdc.cam.ac.uk/data_request/cif. All other data are available from the authors on reasonable request.

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

## Acknowledgements

The present project is supported by CREST, JST. We thank Grants-in-Aid for Scientific Research (nos. JP26288044, JP26105708, JP15K13687 and JP15H05798 to Y.N., no. JP24109014 to K.Y. and no. JP26888008 to H.T.) from JSPS and MEXT. S.K. is a recipient of the JSPS Predoctoral Fellowships for Young Scientists. We also thank the Research Hub for Advanced Nano Characterization at The University of Tokyo for X-ray analysis and EPR spectroscopy. Drs Toshiya Ideue and Masaki Nakano are thanked for the measurement of EPR spectroscopy.

## Author contributions

K.Y. and Y.N. directed and conceived this project. S.K., K.A. and K.N. conducted the experimental work. H.T. and Y.M. conducted the computational work. K.I. conducted the EPR measurements. All authors discussed the results and wrote the manuscript.

## Additional information

**Competing financial interests:** The authors declare no competing financial interests.

