## [Peer review file · Nature Communications]

Reviewers' comments:

Reviewer #1 (Remarks to the Author):

In the work by Yoshizawa, Nishibayashi and coworkers the synthesis, characterisation and catalytic performance with regard to N₂ reduction to NH₃ and N₂H₂ of an iron pincer complex based on the pyrol ligand is presented along with DFT computations aiming at the understanding of the catalytic action of this complex.

In this work previous reportings on catalytic NH₃ formation by the Nishibayashi group using molybdenum pincer complexes are extended and the presentation of this new Fe-N₂ complex and the elucidation of its catalytic action should finally be published in Nature Communications as this work adds very important pieces of information about how catalytic NH₃ formation can occur at a single metal centre. The work is timely and of high general interest to the chemical community. However, there is the need for a thorough revision.

Accordingly, this reviewer RECOMMENDS PUBLICATION AFTER REVISIONS, and this reviewer can accept the paper only, if these corrections are made (see below).

Scientifically the work has been carried out nicely in the experimental part. Also the principal handicraft the authors used for the DFT part is generally acknowledged, however, important recomputations are necessary.

The choice of the density functional and the assessment that led to that choice are made with good reason and ensure the right description of the various spin states, but why didn't the authors switch on Grimme's D3 dispersion correction to also ensure the correct description of nonbonding interactions? I do not recall if the D3 or D3BJ correction was already included in the C.01 revision of Gaussian, however this needs definitely to be done as the authors compute minima and transition state structures of complex molecules interacting with each other. In other words all structures need to be recomputed using a dispersion corrected density functional, otherwise the energies obtained can be STRONGLY misleading. I do not say that they will be wrong, but they can be wrong and this needs to be ensured as very important conclusions are drawn on the basis of the DFT computations.

Furthermore, as these investigations are very important to the field in general, the authors should guarantee a higher level of trust for the computed energies. If I understand correctly, only basis sets of split valence quality were used. This is very common for initial optimizations as the choice of a relatively small basis set saves computation time, but it is NOT state of the art anymore to obtain reliable energies for publication, especially in transition metal catalysis and using charged species. The usage of split-valence energies can be STRONGLY misleading. Again, while this does not necessarily have to be the case one simply needs to make sure, that the energy landscape is described correctly by reoptimizing on the triple-zeta level. Ideally one adds quadruple-zeta single point energies on top of that.

Furthermore, especially when using charged species, the structures need to be reoptimized in the solvent phase. While the authors might have done so, it is not clear from the text in the SI. If solvent inclusion just was done by applying single-point energy calculations, then again, I request reoptimization in the solvent.

Summing this up it is requested to reoptimize all structures in the solvent using a dispersion corrected density functional and basis sets of triple zeta quality (the def2-TZVP basis set of Ahlrichs and coworkers should be a good choice).

While both the experimental and the theoretical parts of this work are by and large well and didactical elegantly written the language needs some polishing. There are a couple of instances where the authors phrase „we believe..." which is very honest on the one hand, but should be removed from a scientific paper. More importantly however, the generally accepted style of how to write down a metal complex should be obeyed, it should read for instance $[\text{FeCl}_2(\text{thf})_{1.5}]$ and not $\text{FeCl}_2(\text{thf})_{1.5}$ (this applies throughout the work).

On page 7, second line of the last paragraph it should read Figure 4 and not Scheme 4.

I am happy to review a revised version again.

Reviewer #2 (Remarks to the Author):

A) Key results: This manuscript describes an approximately square planar terminal N_2 adduct of $\text{Fe}(\text{I})$ that is supported by a bis(phosphine)pyrrole ligand that is able to catalyze at low temperature the reduction of nitrogen directly to ammonia using, and to a much more limited extent hydrazine as byproduct, at low temperature, atmospheric nitrogen, using ether as solvent (and/or THF) and KC_8 and HBArF as the reductant/acid choice. A number of new coordination complexes are prepared and characterized that are reasonably interesting.

B) Originality: The overall conceptual originality is not particularly high, but there is novelty in the creative new iron complexes reported that serve as the catalysts of interest. To elaborate: A number of terminally bonded formal $\text{Fe}(\text{I})$ and $\text{Fe}(\text{0})$ complexes have been shown to give rise to stoichiometric N_2H_4 and NH_3 conversions, and the authors have used analogous conditions reported by Peters et al (as cited in the paper) in several previous reports showing that catalytic yields of NH_3 can be obtained under essentially the same specified conditions as previously reported. Hence, the conditions seem to be key. These authors have likewise used the alternative original Schrock conditions to optimize catalysis using low-talent Mo phosphine complexes and have obtained there very significant turnovers (as high as ~ 30 "fixed" nitrogen molecules in one recent system).

In this paper the authors conclude that the Fe-N_2 anion (formally $\text{Fe}(\text{0})$) is the state that is first protonated to kick off catalysis (rather than the neutral $\text{Fe}(\text{I})$ species). This again is a conclusion by now pretty well documented in the literature for Fe (see for example the cited papers JACS, (2014), pp 1105 for use of the $\text{Fe}(\text{0})\text{-N}_2$ anion for protonation/silylation, and also see IC, (2015), pp 9256 for

a thorough discussion about comparative basicity of an anionic species compared with a neutral species, and also see JACS, (2015), 7803 which shows again that a terminal N₂ anion (in this case formally Fe(-1) is what is productively protonated, and also see Angew (2015), 532 for an L₂Fe system that is only protonated at the L₂Fe-N₂ anion form). The authors should provide more literature context when discussing their specific results/conclusions, not just in this specific instance but more generally, so the new and potentially significant results they report can be distilled from the results/conclusions that are somewhat derivative from what is already known.

The most novel part of the paper in my view is that the authors see some hydrazine formed as a byproduct, and this observation has the potential to be mechanistically interesting and important, though its potential importance is not addressed as well/clearly as it should be. This aspect of the paper could be stronger because it is likely what may most distinguish it conceptually. While the 4-coordinate catalyst structure is novel, 3-coordinate and 5-coordinate catalyst structures based on Fe have been reported. So maybe it shouldn't be so surprising that a 4-coordinate structure might work, too, with phosphine supporting ligands? Still, showing it can is significant, and the authors do this.

c) The approach and the quality of the data is, in general, pretty high. I limit comment to instances where improvement is warranted. One issue of concern is the title catalyst. The authors claim it is a Kramer's doublet, but solution moment is ~3 BM and the low T EPR spectrum looks odd for Fe(I) S = 1/2. For sure, this needs to be sorted and carefully explained because this is the title complex that is the catalyst. It is standard to provide a well-simulated EPR spectrum, and even with substantial spin-orbit coupling, 3 BM for Fe(I) is very atypical. I'm concerned this data may be flawed. Also, the magnetism of the Fe(II) chloride, hydride, and methyl is unusual. They report about 3.7 BM for each of these (solution Evans) but for an S = 1 system one expects about 3 BM, and for an S = 2 system one expects a value closer to 5 BM. Are these values T-dependent? Context from the literature in the maintext would help if one expects such large shifts from the spin-only value based on spin-orbit coupling. According to figure S7 perhaps one should (they show some lit examples there with similarly high moments), but some insight would help (at least me).

These issues are details. A broader concern I have regards the presentation of the data in several areas. Possibly the most impactful observation in relation to the known literature is that N₂H₄ can be generated catalytically and its formation is kinetically competitive with NH₃. But to see a full additional turnover (entry 6 table 1; 2.4 equiv N₂H₄) the authors use THF instead of Et₂O. My understanding is that, even at low temperature, the acid they are using (HBArF) rapidly polymerizes THF. Has this been checked? If so, what is the mechanistic implication as to what might be going on? The authors should cite previous Fe systems (and other systems) that generate N₂H₄.

The authors claim in several places that this Fe system is "the most effective iron-catalyzed nitrogen fixation". See for example the Figure on page 1. These types of statements are not well-founded or valuable. To be specific, the authors report in Table 1 that at 80 equiv KC₈ and 76 equiv HBArF the catalyst affords 4.15 molecules of "fixed" molecules of dinitrogen (according to my math). At lower loadings of KC₈ and HBArF (58 equiv; 48 equiv respectively), under the same exact conditions, Anderson et al established up to 8.5 equiv NH₃ (4.25 "fixed" molecules of nitrogen) per Fe center, with a statistical average of ~7 plus or minus 1 averaged from many runs. In this comparison, the previously reported tris(phosphine)borane system is hence "most effective". But again, this is not a

productive line of argument. The systems appear to be rather similar in efficacy (at least in rough terms), and unless the this paper gets really good statistics (they say average of "multiple" runs, but not how many), one really shouldn't make too many claims about which is specifically better. Indeed, to get higher TONs than those already reported, the authors use 200 equiv KC8 and 184 equiv HBARF - in this case they get 8 "fixed" molecules of N₂. In other words, fourfold increase in substrate (acid/reductant) loading than the previous literature. Hence, to state that this pincer complex is "better" under "these" new conditions, they would need to run a cross-comparison with the tris(phosphine)borane iron catalyst system (and also the CP3 system). My advice is instead is simpler: to not try and claim to much territory because these small increases in fixed N₂ are really just that, rather small increases, and to instead focus on the new novel architecture that gives rise to a new Fe catalyst system (there aren't yet very many and there will likely more to follow) with quite good TONs compared with all the iron systems already reported, and even modestly larger ones in the presence of a vast excess of acid/reductant, but to emphasize that it's unknown how the previously reported catalysts would perform in a direct, apple-to-apple comparison with the present, higher-loading conditions.

That the Fe-H species is catalytically active is interesting and is another novel aspect of this paper.

Theory - I think the theory/mechanistic conclusions in this paper are not well substantiated and are out of place. This communication is a preliminary report, and flooding it with DFT in my view clutters what is sound and reliable. Without good mechanistic experimental data, it just doesn't add much and likely confuses. The theory could be published separately where it would be more properly vetted. I imagine the authors won't agree, but I offer the viewpoint.

D) Statistics could be better (explained at least) and again the authors should take great care in drawing conclusions via apple-to-orange comparisons. They may well not stand up to time.

E) I've hit these points already.

F) Again, I've hit these points.

G) They cite appropriately for the most part, but I think many of the ideas in the paper warrant cross-referencing because conceptually so much is known that is related already. Also, Schrock has shown a Mo-H species can be a catalyst, so a citation is likely warranted.

H) Final suggestion: This is a communication. The authors do a bit too much reviewing of the state of affairs for a communication in my view, and the review doesn't do justice to a great many contributions that precede this work. I think such a review is poorly placed here - would be better placed in a full article. I suggest letting the reported results here stand on their own without overly staking claims about who has published what already and in what order, to provide for a much crisper intro that hits what is really delivered here. Or instead write a full article (which could be equally appropriate) that properly notes that other catalysts for hydrazine evolution have been reported already, and some that do so stoichiometrically, and some that turn N₂H₄ into NH₃, etc... The bottom line is that I think this paper could be and should be more concise, not unlike my review! But generally, I do like the compounds and the chemistry reported, it is well done work in most regards, and my intent is to aid in the paper's improvement by offering all these comments.

Response to Referees' Comments

The followings are our answers to the comments.

As for the comments by Reviewer1

- (1) Reviewer 1 pointed out that "It is requested to reoptimize all structures in the solvent using a dispersion corrected density functional and basis sets of triple zeta quality (the def2-TZVP basis set of Ahlrichs and coworkers should be a good choice)." In the original manuscript, we chose the B3LYP* functional to ensure the proper description of the energy splitting of different spin states, because the experimental information on the ground spin state of the Fe-N₂ complex was available. According to the suggestion, we have also examined the B3LYP-D3 functional, and found that this functional correctly predicts the doublet state to be the ground spin state (see Supplementary Table 10 in the revised version). As a result, along the line with the reviewer's suggestion, we have switched to the B3LYP-D3 functional to ensure the correct description of nonbonding interactions in the protonation reactions studied here. For the basis set employed for optimization, we have examined the def2-TZVP basis set as recommended by the reviewer. However, we found that optimization calculations using this basis set require too much cpu time due to a huge number of the total basis sets (>2000), and therefore we had to give up employing the def2-TZVP basis set for optimization. In the revised manuscript, all intermediates and transition states were reoptimized at the B3LYP-D3/SDD(Fe)&6-31G*(others) level, and then single-point calculations were performed at the B3LYP-D3/def2-TZVP level for obtaining energy profiles of the protonation reactions. Moreover, all the structures were optimized in the solvent phase by using PCM. As presented in Figure 7b, the B3LYP-D3/def2-TZVP results show a qualitatively similar trend as the former B3LYP*/6-311+G** results. Based on the recalculated energy profiles, we can conclude that the reduction of the Fe-N₂ complex would provide a great advantage to the protonation of the N₂ ligand. We really appreciate all of your valuable comments and suggestions for improving the quality of our computational work.
- (2) Reviewer 1 pointed out that "There are a couple of instances where the authors phrase „we believe..." which is very honest on the one hand, but should be removed from a scientific paper". According to the suggestions, we have corrected the indicated words in Abstract on page 1 and line 5 of the third paragraph on page 6 in the revised manuscript. Thank you very much for your valuable suggestion.
- (3) Reviewer 1 pointed out that "More importantly however, the generally accepted style of how to write down a metal complex should be obeyed, it should read for instance [FeCl₂(thf)_{1.5}] and not FeCl₂(thf)_{1.5} (this applies throughout the work)".

According to the suggestions, we have corrected the indicated points in the revised manuscript. Thank you very much for your valuable suggestion.

- (4) Reviewer 1 pointed out that “On page 7, second line of the last paragraph it should read Figure 4 and not Scheme 4”. According to the suggestions, we have corrected the indicated point in the revised manuscript. Thank you very much for your careful reading.

As for the comments by Reviewer2

- (1) Reviewer 2 pointed out that “In this paper the authors conclude that the Fe-N₂ anion (formally Fe(0)) is the state that is first protonated to kick off catalysis (rather than the neutral Fe(I) species). This again is a conclusion by now pretty well documented in the literature for Fe (see for example the cited papers JACS, (2014), pp 1105 for use of the Fe(0)-N₂ anion for protonation/silylation, and also see IC, (2015), pp 9256 for a thorough discussion about comparative basicity of an anionic species compared with a neutral species, and also see JACS, (2015), 7803 which shows again that a terminal N₂ anion (in this case formally Fe(-1) is what is productively protonated, and also see Angew (2015), 532 for an L₂Fe system that is only protonated at the L₂Fe-N₂ anion form). The authors should provide more literature context when discussing their specific results/conclusions, not just in this specific instance but more generally, so the new and potentially significant results they report can be distilled from the results/conclusions that are somewhat derivative from what is already known”. According to the suggestions, we have newly added comments on the indicated references in line 4 of the third paragraph on page 8 in the revised manuscript. Thank you very much for your valuable suggestion.
- (2) Reviewer 2 pointed out that “The most novel part of the paper in my view is that the authors see some hydrazine formed as a byproduct, and this observation has the potential to be mechanistically interesting and important, though its potential importance is not addressed as well/clearly as it should be. This aspect of the paper could be stronger because it is likely what may most distinguish it conceptually”. According to the suggestions, we have modified the Abstract on page 1 in the revised manuscript to emphasize the importance of the hydrazine formation. Thank you very much for your valuable suggestion.
- (3) Reviewer 2 pointed out that “The approach and the quality of the data is, in general, pretty high. I limit comment to instances where improvement is warranted. One issue of concern is the title catalyst. The authors claim it is a Kramer's doublet, but solution moment is ~3 BM and the low T EPR spectrum looks odd for Fe(I) S = 1/2. For sure, this needs to be sorted and carefully explained because this is the title complex that is the catalyst. It is standard to provide a well-simulated EPR spectrum, and even with substantial spin-orbit coupling, 3 BM for Fe(I) is very atypical. I'm concerned this data may be flawed”. According to the suggestions, we have added some comments and references on the magnetic moments of the Fe(I) complex

- 1** in line 14 of the second paragraph on page 4 in the revised manuscript. In addition, we have added the simulated spectrum of **1** in Figure 3 on page 19 in the revised manuscript. We have found that our iron-dinitrogen complex was partially decomposed during the low temperature EPR measurement because our iron complex is highly sensitive and unstable. A broad signal from 2500 G to 3300 G in the low temperature EPR spectrum shown in Supplementary Figure 2 on page S2 in the revised Supplementary Information is considered to be derived from the decomposed species. However, we have reproducibly observed signals at 2500 G ($g = 2.6$) and 3000 G ($g = 2.2$), which are similar to those of the reported Fe(I) complexes with an $S = 1/2$ state (Ohki, Y. et al. *Organometallics* doi: 10.1021/acs.organomet.5b01025). In addition, we have observed no signal at around 1500 G ($g = 4$) in the low temperature EPR spectrum shown in Supplementary Figure 2 on page S2 in the revised Supplementary Information. The Fe(I) complex **1** has a solution magnetic moment of 3 BM, which is larger than the spin only value for an $S = 1/2$ state. However, this value is within the range of the reported Fe(I) complexes with an $S = 1/2$ state (Ohki, Y. et al. *Organometallics* doi: 10.1021/acs.organomet.5b01025 and Deng, L. et al. *Organometallics* doi: 10.1021/acs.organomet.6b00047). As the Reviewer 2 pointed out, we have considered that the large magnetic moments of these complexes may be due to the spin-orbit coupling. Based on the results of the room/low temperature EPR measurements, magnetic moment, and theoretical calculations, we have considered that a spin state of $S = 1/2$. We are now carrying out low temperature EPR and SQUID measurements carefully to investigate magnetic properties of our iron complexes. We will report in details in the next full paper. Thank you very much for your valuable suggestion.
- (4) Reviewer 2 pointed out that “Also, the magnetism of the Fe(II) chloride, hydride, and methyl is unusual. They report about 3.7 BM for each of these (solution Evans) but for an $S = 1$ system one expects about 3 BM, and for an $S = 2$ system one expects a value closer to 5 BM. Are these values T-dependent? Context from the literature in the maintext would help if one expects such large shifts from the spin-only value based on spin-orbit coupling. According to figure S7 perhaps one should (they show some lit examples there with similarly high moments), but some insight would help (at least me). According to the suggestions, we have moved the original comments on the magnetic moments of the Fe(II) complexes **2-4** in Figure S7 in the original Supplementary Information into the second paragraph on page 5 in the revised manuscript. As the Reviewer 2 pointed out, we have considered that the large magnetic moments of these complexes may be due to the spin-orbit coupling. Further measurements of the magnetic property are ongoing and we will report in details in the next full paper. Thank you very much for your valuable suggestion.
- (5) Reviewer 2 pointed out that “Possibly the most impactful observation in relation to the known literature is that N_2H_4 can be generated catalytically and its formation is kinetically competitive with NH_3 . But to see a full additional turnover (entry 6 table 1; 2.4 equiv N_2H_4) the authors use THF instead of Et_2O . My understanding is that,

even at low temperature, the acid they are using (HBArF) rapidly polymerizes THF. Has this been checked? If so, what is the mechanistic implication as to what might be going on?”. When the catalytic reaction was carried out in THF, we have confirmed that no polymeric material was remained after the evaporation of the volatiles. In addition, no polymerization of THF was observed when $[\text{H}(\text{OEt}_2)_2]\text{BAr}^{\text{F}}_4$ was stirred in THF at $-78\text{ }^\circ\text{C}$ for 1h and then at room temperature for 10 min. We have considered that the polymerization of THF by $[\text{H}(\text{OEt}_2)_2]\text{BAr}^{\text{F}}_4$ is negligible under the present reaction conditions. Thank you very much for your valuable suggestion.

- (6) Reviewer 2 pointed out that “The authors should cite previous Fe systems (and other systems) that generate N_2H_4 ”. According to the suggestion, we have added some comments and references (Tyler, D. R. et al. *Coord. Chem. Rev.* **2010**, 254, 1883., Hazari, N. *Chem. Soc. Rev.* **2010**, 39, 4044., Chatt, J. et al. *Chem. Rev.* **1978**, 78, 589., Hidai, M. et al. *Chem. Rev.* **1995**, 95, 1115.) on the iron and other systems that generated hydrazine in line 4 of the second paragraph on page 6 and in line 9 of the second paragraph on page 7 in the revised manuscript. Thank you very much for your valuable suggestion.
- (7) Reviewer 2 pointed out that “The authors claim in several places that this Fe system is “the most effective iron-catalyzed nitrogen fixation”. See for example the Figure on page 1. These types of statements are not well-founded or valuable. To be specific, the authors report in Table 1 that at 80 equiv KC8 and 76 equiv HBArF the catalyst affords 4.15 molecules of “fixed” molecules of dinitrogen (according to my math). At lower loadings of KC8 and HBArF (58 equiv; 48 equiv respectively), under the same exact conditions, Anderson et al established up to 8.5 equiv NH_3 (4.25 “fixed” molecules of nitrogen) per Fe center, with a statistical average of ~ 7 plus or minus 1 averaged from many runs. In this comparison, the previously reported tris(phosphine)borane system is hence “most effective”. But again, this is not a productive line of argument. The systems appear to be rather similar in efficacy (at least in rough terms), and unless the this paper gets really good statistics (they say average of “multiple” runs, but not how many), one really shouldn't make too many claims about which is specifically better. Indeed, to get higher TONs than those already reported, the authors use 200 equiv KC8 and 184 equiv HBArF - in this case they get 8 “fixed” molecules of N_2 . In other words, fourfold increase in substrate (acid/reductant) loading than the previous literature. Hence, to state that this pincer complex is “better” under “these” new conditions, they would need to run a cross-comparison with the tris(phosphine)borane iron catalyst system (and also the CP3 system). My advice is instead is simpler: to not try and claim to much territory because these small increases in fixed N_2 are really just that, rather small increases, and to instead focus on the new novel architecture that gives rise to a new Fe catalyst system (there aren't yet very many and there will likely more to follow) with quite good TONs compared with all the iron systems already reported, and even modestly larger ones in the presence of a vast excess

of acid/reductant, but to emphasize that it's unknown how the previously reported catalysts would perform in a direct, apple-to-apple comparison with the present, higher-loading conditions. According to the suggestions, we have avoided comparing the TONs of Peters' catalysts and ours in the revised manuscript. As a result, we have removed the indicated phrase "the most effective iron-catalyzed nitrogen fixation" in the revised manuscript. Thank you very much for your valuable suggestion.

- (8) Reviewer 2 pointed out that "I think the theory/mechanistic conclusions in this paper are not well substantiated and are out of place. This communication is a preliminary report, and flooding it with DFT in my view clutters what is sound and reliable. Without good mechanistic experimental data, it just doesn't add much and likely confuses. The theory could be published separately where it would be more properly vetted. I imagine the authors won't agree, but I offer the viewpoint". As the Reviewer 2 mentioned, we have considered that the result of the DFT calculations supports the experimental result. In this paper, we have carried out preliminary DFT calculations based on the results of the mechanistic experiments. Of course, we are now doing further experiments to elucidate the detailed reaction mechanism. We will report more detailed results in the near future. Thank you very much for your valuable suggestion.
- (9) Reviewer 2 pointed out that "Statistics could be better (explained at least) and again the authors should take great care in drawing conclusions via apple-to-orange comparisons. They may well not stand up to time". According to the suggestion, we have added some comments on the number of trials in the catalytic reactions in Table 1 in the revised manuscript. In addition, we have removed the comparison between Peters' catalysts and ours in the revised manuscript. Thank you very much for your valuable suggestion.
- (10) Reviewer 2 pointed out that "Schrock has shown a Mo-H species can be a catalyst, so a citation is likely warranted". According to the suggestion, we have added some comments and the indicated reference (Schrock, R. R. et al. *Inorg. Chem.* **2005**, *44*, 1103 as reference 42) in line 12 of the third paragraph on page 6 in the revised manuscript. Thank you very much for your valuable suggestion.
- (11) Reviewer 2 pointed out that "Final suggestion: This is a communication. The authors do a bit too much reviewing of the state of affairs for a communication in my view, and the review doesn't do justice to a great many contributions that precede this work. I think such a review is poorly placed here - would be better placed in a full article. I suggest letting the reported results here stand on their own without overly staking claims about who has published what already and in what order, to provide for a much crisper intro that hits what is really delivered here. Or instead write a full article (which could be equally appropriate) that properly notes that other catalysts for hydrazine evolution have been reported already, and some that do so

stoichiometrically, and some that turn N_2H_4 into NH_3 , etc... The bottom line is that I think this paper could be and should be more concise, not unlike my review! But generally, I do like the compounds and the chemistry reported, it is well done work in most regards, and my intent is to aid in the paper's improvement by offering all these comments". According to the suggestion, we have removed Scheme 1 in the revised manuscript. In addition, we have added some references to the suitable positions in the revised manuscript. Thank you very much for your valuable suggestion.

REVIEWERS' COMMENTS:

Reviewer #1 (Remarks to the Author):

The authors have in their revision provided all necessary information and have undertaken everything that was asked for in my initial report. Therefore, I do now recommend acceptance of the paper. However, one VERY IMPORTANT point is missing: I did not find neither in the MS nor in the SI the part which describes the computational details. This was enclosed in the original version, but is now gone. This MUST be enclosed again. Also in the MS and in the SI the legends for all graphics showing computed data (structures, energy profiles) should be augmented with the information of how that data was computed to make retrieval of that information by the reader easy. If the authors do finalized their manuscripts according to these points the paper can be published.

Reviewer #2 (Remarks to the Author):

This paper is significantly improved and I think is suitable for publication with some minor adjustments. Some final comments I hope will be helpful to the authors are as follows:

The following statement is made: "The present reaction system provides the first example of the catalytic formation of hydrazine as a reduced product directly from molecular dinitrogen by using transition metal-dinitrogen complexes as catalysts."

This statement needs to be carefully qualified - it is perhaps technically true as written (I'd have to dig through all the literature to be certain), but it is a bit misleading(?). Shilov has previously reported thousands of equiv of hydrazine generation catalytically using Mo(III) precursors in the presence of phosphines and reductant in aqueous soln (see for example Table 4 in Coord Chem Rev, 1995, 144 (on pp 98)). The Shilov systems are arguably less well-defined, but given the authors in this paper also cite their use of simple iron precursors in the presence of strong reductant and silyl electrophiles to afford catalytic nitrogen fixation to tris(trimethyl)silylamine, I would think they would agree this early Shilov work is worth noting in the context of catalytic hydrazine generation using soluble transition metal systems, even if a molecular M-N₂ complex was not used as the pre-catalyst, but rather a metal halide.

Pertaining to the following statement: "This result is in sharp contrast to the reactivity of iron-hydride complexes reported by Peters and co-workers, where these iron-hydride complexes did not work as catalysts at all."

This is not a true statement. In the 2013 report by Peters and coworkers, the hydride was able to produce some ammonia, but not catalytic yields. In a more recent study, this hydride is shown to be catalytically competent when better solubilized, and has also been identified via in situ spectroscopy as a catalyst resting state: See J. Am. Chem. Soc., 2016, 138 (16), pp 5341-5350. The authors should adjust their wording here for accuracy.

The authors may also want to cite (?) an additional paper consistent with the following statement:

"In fact, the alternating pathway has been proposed for the stoichiometric formation of ammonia and hydrazine from iron-dinitrogen complexes based on the reactivity of isolated and generated intermediates". See *J. Am. Chem. Soc.*, 2016, 138 (12), pp 4243.

Responses to Reviewers' comments (NCOMMS-16-01717-A)

As for the comments by Reviewer1

- (1) Reviewer 1 pointed out that "The authors have in their revision provided all necessary information and have undertaken everything that was asked for in my initial report. Therefore, I do now recommend acceptance of the paper. However, one VERY IMPORTANT point is missing: I did not find neither in the MS nor in the SI the part which describes the computational details. This was enclosed in the original version, but is now gone. This MUST be enclosed again." We consider that this reviewer has overlooked the detailed computational method on pages S91-S92 in the revised Supplementary Information. Thank you very much for your valuable suggestion.
- (2) Reviewer 1 pointed out that "Also in the MS and in the SI the legends for all graphics showing computed data (structures, energy profiles) should be augmented with the information of how that data was computed to make retrieval of that information by the reader easy". As described in the above as for the comments by Reviewer 1, we have described enough information on the detailed computational method on pages S91-S92 in the revised Supplementary Information. To avoid the duplication of the detailed computational method in the present manuscript, we have not newly added the detailed computational method in the legend for all graphics. Thank you very much for your valuable suggestion.

As for the comments by Reviewer2

- (1) Reviewer 2 pointed out that "This statement needs to be carefully qualified - it is perhaps technically true as written (I'd have to dig through all the literature to be certain), but it is a bit misleading(?). Shilov has previously reported thousands of equiv of hydrazine generation catalytically using Mo(III) precursors in the presence of phosphines and reductant in aqueous soln (see for example Table 4 in *Coord Chem Rev*, 1995, 144 (on pp 98). The Shilov systems are arguably less well-defined, but given the authors in this paper also cite their use of simple iron precursors in the presence of strong reductant and silyl electrophiles to afford catalytic nitrogen fixation to tris(trimethyl)silylamine, I would think they would agree this early Shilov work is worth noting in the context of catalytic hydrazine generation using soluble transition metal systems, even if a molecular M-N₂ complex was not used as the pre-catalyst, but rather a metal halide". According to the suggestions, we have newly added comments on the indicated references (Shilov, A. E. *Coord. Chem. Rev.* 1995, 144, 69; Shilov, A. E. *Russ. Chem. Bull.* 2003, 52, 2555) in line 5 of the Discussion part on page 11 in the re-revised manuscript. Thank you very much for your valuable suggestion.
- (2) Reviewer 2 pointed out that "Pertaining to the following statement: "This result is in sharp contrast to the reactivity of iron-hydride complexes reported by Peters and co-workers, where these iron-hydride complexes did not work as catalysts at all." This is not a true statement. In the 2013 report by Peters and coworkers, the hydride was able to produce some ammonia, but not catalytic yields. In a more recent study, this hydride is shown to be catalytically competent when better solubilized, and has also been identified via in situ spectroscopy as a catalyst resting state: See *J. Am. Chem. Soc.*, 2016, 138 (16), pp 5341-5350. The authors should adjust their wording

here for accuracy". According to the suggestions, we have added some comments and the indicated reference (Peters, J. C. *J. Am. Chem. Soc.* 2016, 138, 5341) in line 15 of the third paragraph on page 6 in the re-revised manuscript. Thank you very much for your valuable suggestion.

- (3) Reviewer 2 pointed out that "The authors may also want to cite (?) an additional paper consistent with the following statement: "In fact, the alternating pathway has been proposed for the stoichiometric formation of ammonia and hydrazine from iron-dinitrogen complexes based on the reactivity of isolated and generated intermediates". See *J. Am. Chem. Soc.*, 2016, 138 (12), pp 4243". According to the suggestions, we have added some comments and the indicated reference (Peters, J. C. *J. Am. Chem. Soc.* 2016, 138, 4243) in the last sentence of the second paragraph on page 7 in the re-revised manuscript. Thank you very much for your valuable suggestion.